# Production of Proinflammatory Cytokines by CD4+ and CD8+ T Cells in Response to Mycobacterial Antigens among Children and Adults with Tuberculosis

**DOI:** 10.3390/pathogens12111353

**Published:** 2023-11-14

**Authors:** Erin Morrow, Qijia Liu, Sarah Kiguli, Gwendolyn Swarbrick, Mary Nsereko, Megan D. Null, Meghan Cansler, Harriet Mayanja-Kizza, W. Henry Boom, Phalkun Chheng, Melissa R. Nyendak, David M. Lewinsohn, Deborah A. Lewinsohn, Christina L. Lancioni

**Affiliations:** 1School of Medicine, Oregon Health and Science University, Portland, OR 97239, USA; 2School of Public Health, Oregon Health and Science University, Portland, OR 97239, USA; 3Department of Pediatrics, Makerere University, Mulago Hill Road, Kampala P.O. Box 7072, Uganda; 4Department of Pediatrics, Oregon Health and Science University, Portland, OR 97239, USA; 5Uganda-Case Western Research Collaboration, Case Western Reserve University, Cleveland, OH 44106, USA; 6Department of Medicine, Makerere University, Mulago Hill Road, Kampala P.O. Box 7072, Uganda; 7Department of Medicine, Case Western Reserve University, Cleveland, OH 44106, USA; 8Department of Medicine, Oregon Health and Science University, Portland, OR 97239, USA; 9Division of Pulmonary and Critical Care Medicine, Portland VA Medical Center, Portland, OR 97239, USA

**Keywords:** pediatric, *Mycobacterium tuberculosis*, adaptive immunity, cytokines, T cells

## Abstract

Tuberculosis (TB), caused by *Mycobacterium tuberculosis* (Mtb), remains a leading cause of pediatric morbidity and mortality. Young children are at high risk of TB following Mtb exposure, and this vulnerability is secondary to insufficient host immunity during early life. Our primary objective was to compare CD4+ and CD8+ T-cell production of proinflammatory cytokines IFN-gamma, IL-2, and TNF-alpha in response to six mycobacterial antigens and superantigen staphylococcal enterotoxin B (SEB) between Ugandan adults with confirmed TB (n = 41) and young Ugandan children with confirmed (n = 12) and unconfirmed TB (n = 41), as well as non-TB lower respiratory tract infection (n = 39). Flow cytometry was utilized to identify and quantify CD4+ and CD8+ T-cell cytokine production in response to each mycobacterial antigen and SEB. We found that the frequency of CD4+ and CD8+ T-cell production of cytokines in response to SEB was reduced in all pediatric cohorts when compared to adults. However, T-cell responses to Mtb-specific antigens ESAT6 and CFP10 were equivalent between children and adults with confirmed TB. In contrast, cytokine production in response to ESAT6 and CFP10 was limited in children with unconfirmed TB and absent in children with non-TB lower respiratory tract infection. Of the five additional mycobacterial antigens tested, PE3 and PPE15 were broadly recognized regardless of TB disease classification and age. Children with confirmed TB exhibited robust proinflammatory CD4+ and CD8+ T-cell responses to Mtb-specific antigens prior to the initiation of TB treatment. Our findings suggest that adaptive proinflammatory immune responses to Mtb, characterized by T-cell production of IFN-gamma, IL-2, and TNF-alpha, are not impaired during early life.

## 1. Introduction

Tuberculosis disease (TB) develops following infection by *Mycobacterium tuberculosis* (Mtb). Despite decades of public health interventions to control this disease, including wide-spread use of Bacillus Calmette–Guérin (BCG) vaccine during early infancy, Mtb remains the leading driver of infection-related morbidity and mortality worldwide [1]. Among children, there are over one million cases of pediatric TB reported annually and nearly a quarter of affected children die from the disease [2]. Given the challenges of confirming the diagnosis of TB in young children, the burden of pediatric TB is thought to be dramatically underestimated [3].

Mtb is spread from person-to-person by airborne transmission by individuals with TB who aerosolize the highly contagious mycobacterium when coughing [4]. Following exposure and infection with Mtb, the organism is capable of establishing an asymptomatic, latent infection within the lung and associated lymph nodes, or may progress to an active infection and symptomatic TB. Although an otherwise healthy adult has only a 5–10% risk of developing TB following Mtb infection over their entire lifetime, the risk of TB is far greater among people living with HIV, as well as children under 5 years old regardless of HIV status. Indeed, HIV-uninfected infants under 1 year old have a 30–40% chance of developing pulmonary TB within 12 months of Mtb infection [5]. Moreover, young children are highly vulnerable to more severe clinical manifestations of TB, such as TB meningitis and miliary or disseminated TB, that have mortality rates as high as 20% [6]. Although the increased vulnerability to TB has long been recognized in clinical and epidemiologic studies [7], the biologic mechanisms responsible for this age-associated risk in early childhood remain poorly understood.

CD4+ T cells, cellular components of the adaptive immune system, are essential to host defense against progression from Mtb infection to TB disease [8]. Inherent, age-related differences in adaptive immune function may contribute to the increased risk of TB disease following Mtb exposure that is observed during early childhood. For example, the cellular components of the adaptive arm of the immune system, such as CD4+ and CD8+ T cells, exhibit a naïve phenotype during infancy. These naïve T cells must receive more co-stimulation to become activated, effector cells, when compared to memory T cells [9,10]. Using in vitro models of CD4+ T-cell activation where cells were stimulated through the T-cell receptor and provided with co-stimulation through CD28, CD4+ T cells isolated from neonates and young infants demonstrated limited capacity to produce the prototypical proinflammatory Type-1 helper T cell (Th-1) cytokine interferon-gamma (IFN-γ). Rather, in these models, infant CD4+ T cells produced IL-4 and the anti-inflammatory cytokine IL-10 [11,12,13,14,15,16,17]. Notably, IL-10 is considered a regulatory cytokine, and has been associated with the capacity of Mtb to evade immunity and establish disease [18]. Young infants are also known to have an expanded pool of anti-inflammatory regulatory CD4+ T cells which produce transforming growth factor-beta (TGF-β) that enhances suppressive function during infancy. The expansion of regulatory CD4+ T cells has been demonstrated among children with TB, although it is unclear if this expansion contributes to vulnerability to TB or is a response to the inflammation associated with active TB [19,20,21].

CD8+ T cells have also emerged as important contributors to host defense following Mtb-infection [22]. Similar to Th-1 type CD4+ T cells, CD8+ T cells can produce proinflammatory cytokines IFN-γ, interleukin-2 (IL-2) and tumor necrosis factor-alpha (TNF-α) following activation, but also have cytotoxic capacity through the production of molecules such as perforin and granulysin. In addition, CD8+ T cells recognize intracellular pathogens such as Mtb, and have been shown to lyse cells heavily infected with Mtb [23]. Similar to CD4+ T cells, studies of CD8+ T cell phenotype and function during infancy and early childhood demonstrated their unique features when compared to CD8+ T cells from adults. For example, neonatal CD8+ T cells are less likely to express IFN-γ, and to produce multiple cytokines following activation [24].

The bias against CD4+ Th-1, and CD8+ T-cell proinflammatory responses during infancy has been primarily shown using polyclonal or mitogen-based stimulation conditions. In contrast, numerous studies have demonstrated that T cells from infants vaccinated with the live BCG vaccine, as well as infants and young children with asymptomatic Mtb infection and those ill with TB, are capable of mounting Mtb-specific CD4+ and CD8+ T-cell responses characterized by robust IFN-γ production [25,26,27,28,29,30,31,32]. Understanding if T-cell proinflammatory cytokine responses to Mtb are compromised in young children is critical to understanding the immunobiology of pediatric TB, and represents a critical knowledge gap as this information is vital to the development of novel strategies (such as next-generation vaccines) to prevent TB in this vulnerable population. Given our prior studies of Mtb-specific T-cell responses in young children have demonstrated robust CD4+ and CD8+ T-cell proinflammatory responses [25,30], we hypothesized that T cells from young children with TB would have a similar capacity to produce proinflammatory cytokines when compared to T cells from adults with TB.

Here, our primary objective was to probe the capacity of Mtb-specific CD4+ and CD8+ T cells from HIV-uninfected Ugandan infants and young children with confirmed TB to produce three key proinflammatory cytokines (IFN-γ, IL-2, and TNF-α) in response to the well-characterized Mtb-specific antigens ESAT-6 and CFP-10, as well as five mycobacterial antigens previously shown to be immunodominant CD8+ T-cell antigens among Mtb-infected adults [33]. As a reference, we also measured T-cell cytokine responses to the T-cell superantigen Staphylococcal enterotoxin B (SEB), that has previously been shown to elicit reduced proinflammatory responses in young children as compared to adults [34,35]. We compared cytokine production from Ugandan children with confirmed TB to that elicited from Ugandan adults with confirmed TB, as well as age-matched Ugandan children with non-TB lower respiratory tract infection (non-TB LRTI) and unconfirmed TB. Importantly, our findings support that T cells from young children with confirmed TB exhibit an adult-like capacity to produce proinflammatory cytokines in response to Mtb-specific antigens, whereas children with unconfirmed TB demonstrate limited T-cell responses that likely reflect the diagnostic uncertainly of TB disease despite their clinical presentation consistent with this challenging diagnosis. We also demonstrate that T cells from Ugandan adults and children exhibit distinctive patterns of cytokine production in response to antigens shared among Mtb and non-TB mycobacterial antigens, an important consideration for next-generation vaccines against TB disease in young children.

## 2. Materials and Methods

### 2.1. Study Setting, Design, and Population

Children 1–60 months old hospitalized in the pediatric wards of Mulago Hospital in Kampala, Uganda, between 2011 and 2014, with an admission diagnosis of severe pneumonia, were eligible to participate. Children with a prior diagnosis of TB or children currently on TB treatment for 7 or more days, were excluded. All children enrolled in this study underwent a diagnostic evaluation for TB disease, including chest X-ray (CXR), tuberculin skin test (TST), and one sputum induction to obtain materials for acid-fast bacillus (AFB) smear, liquid, and solid culture (Middlebrook media). Growth of Mtb on AFB culture was confirmed using PCR. All children underwent diagnostic testing for HIV infection using a rapid-diagnostic assay; infants < 18 months were also tested for HIV using HIV-DNA PCR assay. All children underwent a single blood draw to obtain research materials for immunologic assays, and were followed up by research clinicians 2 months following hospital discharge to assess response to therapy. Inpatient clinical care and treatment decisions were guided by hospital physicians following routine clinical protocols. CXRs were interpreted using a standardized research tool by a radiologist well experienced in the evaluation of pediatric and adult TB suspects. Final research classifications of confirmed TB, unconfirmed TB, and non-TB LRTI were performed by consensus review by three study physicians (CLL; DAL; SK) after completion of 2 month follow-up visit and finalization of all microbiologic cultures. Standardized definitions for confirmed and unconfirmed intrathoracic TB were applied [36]. Confirmed TB was defined as isolation of Mtb from an induced sputum sample by AFB culture in a child with signs/symptoms of TB disease, as defined below. Unconfirmed TB was defined as presence of at least 2 of the following (in absence of bacteriologic confirmation): CXR consistent with pulmonary TB; clinical signs/symptoms suggestive of TB including persistent cough of 2 weeks duration or longer, failure to thrive, unexplained fever for 1 week or longer, persistent lethargy, known close contact with adult with confirmed TB, positive TST, or clinical response to TB treatment. Non-TB LRTI was defined by all of the following among children hospitalized with severe pneumonia: absence of bacteriologic confirmation of TB, child did not demonstrate 2 or more features of unconfirmed TB, negative TST, no known TB exposure, and clinical improvement in the absence of TB treatment. For this analysis, only HIV-uninfected children were included.

Ugandan adults, ages 18–65 years old, were recruited from the National TB Treatment Center at Mulago Hospital between 2011–2014. All adults underwent CXR and collection of 3 expectorated sputum samples for AFB smear and culture; growth of Mtb on AFB culture was confirmed using PCR. All adults were tested for HIV using a rapid diagnostic assay. Only HIV-uninfected adults with culture-confirmed pulmonary TB were eligible for this analysis.

### 2.2. Blood Sample Collection, Processing, and Storage

Whole blood was collected from all study participants prior to placement of TST (in children) and initiation of TB treatment. For children, up to 4 mL of whole blood was collected in CPT tubes (BD Biosciences), and peripheral blood mononuclear cells (PBMC) isolated using sterile technique following a standardized protocol [37]. Among adults, PBMC were isolated from whole blood by density gradient centrifugation method using Ficoll Histopaque, as previously reported [33,38]. All PBMC were cryopreserved in liquid nitrogen (−140 °C) for batch analysis, as detailed below.

### 2.3. T-Cell Stimulation Assay and Flow Cytometry

A standardized protocol previously applied in our laboratory [38] and adopted from published protocols [39] was applied to detect cytokine production by CD4+ and CD8+ T cells. Cryopreserved PBMC were thawed at 37 °C, and rested overnight at 37 °C/5% CO_2_ in X-VIVO serum-free media. Viable cells were then counted and incubated at 37 °C/5% CO_2_ in X-VIVO media for 18 h in 96-well tissue culture plates (2 million PBMC per well) with individual peptide pools consisting of 15-mer peptides overlapping by 11 amino acids (10 μg/mL) representing: ESAT6/CFP10, the EsxJ family, PE12:PE13, PE3, PPE15, and PPE51 (Table 1). PBMC incubated with media alone served to capture non-specific (resting/background) levels of cytokine production. PBMC incubated with Staphylococcal enterotoxin B (SEB; 1 μg/mL) served as a positive control. FastImmune Co-Stimulatory Antibodies, (CD28/CD49d, BD) were added to all wells (200 ng/mL). The protein transport inhibitor Brefeldin A (Sigma-Aldrich, St. Louis, MO, USA), was added during the final 12 h of incubation to optimize detection of intracellular cytokines. Following 18 h of incubation, cells were washed with PBS in tissue culture plates and stained with LIVE/DEAD Violet indicator, followed by staining in cold PBS with monoclonal antibodies for: CD4 PeCy7 and CD8 APC H7. Cells were subsequently treated with fixation/permeabilization solution (BD Biosciences Cytofix/Cytoperm, San Jose, CA, USA) and stained with monoclonal antibodies for CD3 PerCP (stained intracellularly to optimize detection of activated T cells), IFN-γ FITC, IL-2 APC, and TNF-α PE. Cellular populations were subsequently acquired using a MACSQuant Flow cytometer (Miltenyi Biotec, Bergisch Gladbach, North Rhine-Westphalia, Germany), and analyzed using FlowJo software version 10.1 (BD). CD3+CD4+ and CD3+CD8+ T-cell production of intracellular cytokines was determined using a standardized gating scheme. To reduce bias in gating, all analysis was performed by a single investigator who was blinded to the participant cohort. To ensure that results were reproducible among experiments, PBMC from a single, healthy laboratory donor were included in each experiment. Representative data are shown in Appendix A.

### 2.4. Statistical Analysis

CD3+CD4+ and CD3+CD8+ T-cell responses were considered separately. Adjusted frequency (raw frequency of cytokine production following stimulation minus the frequency of cytokine production in resting/unstimulated conditions), was used to account for the potential presence of background, or non-specific, responses. The overall proportions of children and adults with a positive cytokine response (defined as ≥0.05% of CD3+CD4+ or CD3+CD8+ T cells producing a cytokine following background correction) to each peptide pool were compared using Fisher-Exact test. Post hoc pairwise comparisons using the Fisher’s exact test with adjusted *p*-values (Bonferroni correction) were conducted. The median percentages of CD3+CD4+ and CD3+CD8+ T cells with cytokine responses to each peptide pool (following background correction) were compared by Kruskal–Wallis test. Post hoc comparisons were performed using Dunn’s test of multiple comparisons. The Fisher’s exact test, Kruskal–Wallis test, and Dunn’s test were selected as statistical methods for analysis due to limited sample size. Adjusted (Bonferroni correction) *p*-values < 0.05 were considered significant.

### 2.5. Ethics Approval and Consent to Participate

This study received Institutional Review Board approval from all sponsoring institutions. Written, informed consent was obtained from all participants prior to study enrollment in their preferred language. For participating children, written informed consent was obtained from a parent or guardian.

## 3. Results

### 3.1. Participant Recruitment

A total of 92 HIV-uninfected Ugandan children < 5 years old hospitalized with severe pneumonia recruited from the pediatric wards of Mulago Hospital in Kampala, Uganda, were included in this analysis. Of the 92 children, 12 were classified with confirmed pulmonary TB, 41 unconfirmed pulmonary TB, and 39 non-TB pediatric LRTI. Forty-one HIV-uninfected Ugandan adults with culture-confirmed pulmonary TB were recruited from outpatient adult TB treatment centers in Kampala, Uganda. All pediatric and adult participants underwent a single blood draw to obtain research materials prior to initiation of TB treatment. The demographic characteristics of all study participants are shown in Table 2.

### 3.2. CD4+ and CD8+ T Cells from Young Children with Confirmed TB Exhibit Adult-Equivalent Cytokine Responses to Mtb-Specific Antigens ESAT-6 and CFP-10

Intracellular cytokine (IFN-γ, IL-2, and TNF-α) production by CD4+ and CD8+ T cells from Ugandan children with confirmed TB, unconfirmed TB, non-TB LRTI, and adults with confirmed TB, in response to Mtb-specific antigens ESAT-6 and CFP-10 (EC) was assessed by flow cytometry (Appendix A). When comparing EC-responsive CD4+ and CD8+ T-cell responses among cohorts, an equivalent proportion of children and adults with confirmed TB exhibited IFN-γ, IL-2, and TNF-α production. However, a significantly lower proportion of children with unconfirmed TB exhibited CD4+ T-cell production of IFN-γ and TNF-α in response to EC when compared to either children or adults with confirmed disease. The portion of children with unconfirmed TB who demonstrated CD4+ T-cell production of IL-2 was also significantly reduced when compared to children with confirmed disease. Similarly, children with unconfirmed TB were less likely to have CD8+ T cells productive of IFN-γ, IL-2 and TNF-α, as compared to adults and children with confirmed disease. Children classified as non-TB LRTI did not exhibit T-cell cytokine responses to EC (Figure 1).

When examining the median percentages of CD4+ and CD8+ T cells that produced cytokines in response to EC-peptide pools and the pattern of cytokine production, children and adults with confirmed disease had equivalent medians and patterns of multiple cytokine production (Figure 2; Appendix A). Here again, the median response and pattern of cytokine production by EC-specific CD4+ and CD8+ T cells from children with unconfirmed TB were significantly different when compared to both children and adults with confirmed TB.

### 3.3. Mycobacterial Antigens PE3 and PPE15 Are Highly Immunogenic but Elicit Unique Responses between Ugandan Children and Adults with Confirmed TB

Intracellular cytokine (IFN-γ, IL-2, and TNF-α) production by CD4+ and CD8+ T cells from Ugandan children with confirmed TB, unconfirmed TB, non-TB LRTI, and adults with confirmed TB, in response to five different mycobacterial antigens previously shown to be immunogenic in Mtb-infected adults [33], was assessed by flow cytometry (Table 1). Here, the proportion of children and adults with CD4+ T-cell cytokine responses to each antigen was highly variable with notable differences in responses to PE3 and PPE15 between children and adults with confirmed TB (Figure 1). The proportion of children and adults with confirmed TB who exhibited CD8+ T-cell cytokine responses to PE3 and PPE 15 were similar; moreover, both antigens were commonly recognized in children with unconfirmed TB and non-TB LRTI (Figure 1). These antigens are not specific to Mtb, but found in other mycobacteria and BCG-vaccine preparations. The median percentages of CD4+ T cells that produced cytokines, and patterns of cytokine production by CD4+ T cells responsive to PE3 and PPE15, were also different between children and adults with confirmed TB, as well as children with confirmed TB compared to those with non-TB LRTI. The median percentages of CD8+ T cells, and patterns of cytokine production, in response to PE3 and PPE15 was similar across all groups (Figure 3 and Figure 4; Appendix A).

### 3.4. Low Immunogenicity of Mycobacterial Antigens EsxJ, PE12/13, and PPE51 among Children with Unconfirmed TB and non-TB LRTI

CD4+ T-cell responses to PE12/13 and PPE51 peptide pools were similar in both children and adults with confirmed TB. There was a notable trend that a higher proportion of adults with confirmed TB demonstrated a CD8+ IFN-γ response to PE12/13 than was observed among the pediatric cohorts. The proportion of children with unconfirmed TB and non-TB LRTI with a detectable CD4+ or CD8+ T-cell response to PE12/13 and PPE51 was extremely low (Figure 1; Appendix A). T-cell responses to EsxJ were reduced in all groups as compared to the other antigens evaluated.

### 3.5. T-Cell Production of Cytokines in Response to the Superantigen SEB Is Reduced among Ugandan Children as Compared to Adults

The capacity of CD4+ and CD8+ T cells to produce proinflammatory cytokines to superantigen SEB was compared among Ugandan children with confirmed TB, unconfirmed TB, non-TB LRTI, and adults with confirmed TB. Although the proportion of individuals who exhibited T-cell production of proinflammatory cytokines was not different among the groups (data not shown), the median percentages of CD4+ and CD8+ T cells productive of cytokines were significantly different. Here, the median percentages of CD4+ T cells productive of IFN-γ and IL-2 were reduced among Ugandan children from all cohorts when compared to adult responses (Figure 5). CD8+ T-cell IL-2 production was also reduced in the Ugandan children from all cohorts when compared to adult responses. Children with unconfirmed TB exhibited reduced CD8+ T cell production of IFN-γ and TNF-α as compared to adults.

## 4. Discussion

Infants and young children are uniquely vulnerable to developing TB disease quickly following Mtb-exposure, and to experiencing more severe disease manifestations such as TB meningitis and disseminated TB when compared to immunocompetent adults [5,40]. The underlying immunobiology of TB disease in young hosts remains poorly understood. Although the contribution of T-cell production of proinflammatory cytokines IFN-γ and TNF-α toward protective immunity against Mtb has become controversial [41,42,43,44,45], the capacity of T cells from acutely ill children with and without TB to produce this type of cytokine response has not been comprehensively studied. Here, we worked with cohorts of young, hospitalized, Ugandan children with confirmed and unconfirmed TB, as well as non-TB LRTI, to compare the expression of proinflammatory cytokines IFN-γ, IL-2, and TNF-α by mycobacterial-reactive CD4+ and CD8+ T cells, to those observed among Ugandan adults with confirmed TB. Our findings demonstrate that CD4+ and CD8+ proinflammatory cytokine responses to Mtb-specific antigens EC among young children with confirmed disease are equivalent to those observed in adults. Our results support prior reports that IFN-γ production in response to BCG and other mycobacterial antigens is similar in infants, older children, and adults [25,26,28]. Of the six mycobacterial antigens tested, EC is the only set of antigens uniquely expressed by members of the Mycobacterium tuberculosis complex. Our findings add to the emerging body of literature that young children are capable of mounting vigorous, proinflammatory CD4+ and CD8+ T-cell responses against Mtb [25,26,30].

In contrast to the robust proinflammatory cytokine responses to EC that were observed among children and adults with confirmed TB, a minority of children with unconfirmed TB demonstrated proinflammatory responses to these antigens. It is possible that children with unconfirmed TB were more immunosuppressed than those with confirmed disease, thus limiting EC responses. This explanation is unlikely, however, as children with unconfirmed TB exhibited robust proinflammatory responses to other mycobacterial antigens, particularly CD8+ IFN-γ and TNF-α responses to PE3 and PPE15. Children with unconfirmed TB may also have generated an alternative, non-Th1-type response to EC-peptide pools that was not captured in our study. An alternative explanation is that a significant proportion of children classified as unconfirmed TB, despite fulfilling international diagnostic criteria for pediatric TB [36] and receiving treatment for TB based on independent assessment by experienced clinicians, did not have TB disease. The limitations of the currently applied criteria for unconfirmed TB in children are well recognized [46,47]. Despite lacking specificity for pulmonary TB, these guidelines are meant to reduce the number of missed TB diagnoses in young children, given the high risk of severe TB disease during early life. In research settings, children with confirmed and unconfirmed TB are often considered as a single group in order to increase the power of statistical analyses. Here, however, combining results from these two pediatric groups for comparison would have dramatically altered our results and conclusions, with pediatric cytokine responses to EC appearing significantly less than those observed in adults. This finding should be considered for other studies examining pediatric immune responses to Mtb, where understanding the immunobiology of TB disease (or protection against TB disease) is a primary outcome. Fortunately, novel non-sputum-based approaches to confirm TB in young children show promise, and will allow more precise case definitions to be applied to future studies [48,49].

Our study was distinct as we included young Ugandan children hospitalized with non-TB LRTI, an approach that allowed us to assess the immunogenicity of a panel of mycobacterial antigens among acutely ill children with negative evaluation for TB disease, and without evidence of immunologic sensitization to Mtb as determined by negative TST. Our findings confirmed the immunogenicity of mycobacterial antigens PE3 and PPE15, that were broadly recognized by Ugandan children and adults, but lacked specificity for TB disease given the robust responses observed among children in the non-TB LRTI cohort. Notably, CD4+ T-cell cytokine responses to these antigens differed between children and adults, with predominately IL-2 versus IFN-γ production, respectively. The propensity to produce IL-2 may be related to an increased predominance of naïve T cells among young children, although detailed T-cell phenotypes were not assessed in our study. Both antigens were widely recognized by CD8+ T cells resulting in IFN-γ and TNF-α production from both age groups, a finding that reflects their initial discovery as immunodominant-CD8+ T-cell antigens among adults [33]. Our study was not designed to identify antigens for next-generation vaccines against Mtb. However, novel mycobacterial antigens do have the potential to contribute to improvement of the existing BCG vaccine or the creation of a new TB vaccine, in line with the WHO goal to end TB by 2035 [50]. The differences in T-cell cytokine responses to our novel mycobacterial antigens between young children and adults underscore the complexity of vaccination development, and the importance of performing immunogenicity and efficacy studies in a target population living in TB-endemic settings.

Our study incorporated the super-antigen SEB as a positive control in our T-cell stimulation assay and provided interesting results. SEB is a T-cell mitogen capable of broadly binding cell-surface receptors located at the immunologic synapse between antigen-presenting cells and T cells to elicit proinflammatory cytokine production, as well as T-cell proliferation [51]. Although SEB-induced T-cell activation does vary among T cells isolated from healthy adults, at least 20% of T cells are expected to produce a proinflammatory cytokine response. Optimal SEB-induced T-cell activation is dependent on the expression of cellular co-stimulatory molecules such as CD28 [52]. As previously reported [34,35], the median percentages of T cells that generated proinflammatory T-cell responses to SEB were reduced in children with confirmed TB as compared to adults. This is notable, as in the same participants, the median percentages of T-cell responses to EC were equivalent. This finding suggests that T-cell responses to Mtb-specific antigens were preserved in the context of an active mycobacterial infection in young children with confirmed TB. Our findings are supportive of previously published studies demonstrating robust proinflammatory responses to mycobacterial antigens across the age span [25,26,27,28,30]. We have also shown that the frequency of CD4+ and CD8+ T-cell responses to SEB did not differ across the three pediatric cohorts, supporting that within this age group, the capacity to mount proinflammatory T-cell cytokine responses to a superantigen was not intrinsically different based on TB-disease classification.

Our study has several limitations. Firstly, the number of children with confirmed TB was limited; only one induced sputum sample was tested among children, and more modern tests to confirm TB disease (such as Xpert^®^ MTB/RIF Ultra) were not included. Thus, there may have been children within the unconfirmed TB cohort with confirmed disease if a more sensitive microbiologic evaluation was available. Despite this small sample size, however, our statistical findings remained robust even following correction for multiple comparisons. Second, our study only examined IFN-γ, IL-2, and TNF-α production, and did not consider differences in the production of regulatory cytokines such as IL-10 or TGF-β that have been associated with TB disease progression [18,21]. In addition, our study design captured children and adults already exhibiting symptomatic TB (or non-TB LRTI), and thus we cannot determine if T-cell production of these cytokines differed prior to disease onset, or was associated with risk of disease following Mtb infection in different age groups. Asymptomatic children with evidence of Mtb infection were not included as a comparator group. However, a particular strength of our study was the inclusion of acutely ill children from the same community with non-TB LRTI. Access to this population is particularly important when considering T-cell responses to mycobacterial antigens present in both BCG and environmental mycobacteria, and to assess the impact of acute illness on T-cell proinflammatory cytokine production among young children with both TB- and non-TB LRTI.

In conclusion, acutely ill, young children with confirmed TB are capable of producing adult-like proinflammatory cytokine responses to Mtb-specific antigens, despite their reduced responses to the superantigen SEB. In contrast, proinflammatory cytokine responses to Mtb-specific antigens among children with unconfirmed TB differed from both adults and children with confirmed disease. Robust CD4+ T cell IL-2 and CD8+ T cell IFN-γ and TNF-α production in response to PPE15 and PE3 among all pediatric cohorts further supports that CD4+ and CD8+ T cells from young children are capable of recognizing mycobacterial antigens and generating proinflammatory immune responses.

## Figures and Tables

**Figure 1 pathogens-12-01353-f001:**
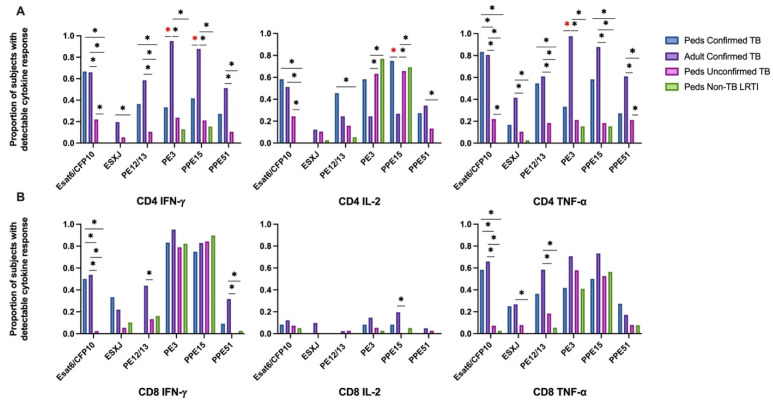
**Proportion of children and adults with detectable CD4+ and CD8+ T-cell cytokine response to six mycobacterial antigens.** PBMC isolated from children with confirmed (n = 12), unconfirmed TB (n = 41), non-TB lower respiratory tract infection (n = 39) and adults with confirmed TB (n = 41), were thawed in batches, rested overnight, and CD4+ and CD8+ T-cell production of IFN-γ, IL-2, and TNF-α in response to 6 peptide pools representing mycobacterial antigens quantified by intracellular flow cytometry following 18 h of stimulation (ICS). A positive response to each peptide pool was defined as ≥0.05% of CD4+ (**A**) or CD8+ (**B**) T cells producing a cytokine following background correction. * Indicates adjusted *p*-value ≤ 0.05. * Illustrates significant differences between adults and children with confirmed TB.

**Figure 2 pathogens-12-01353-f002:**
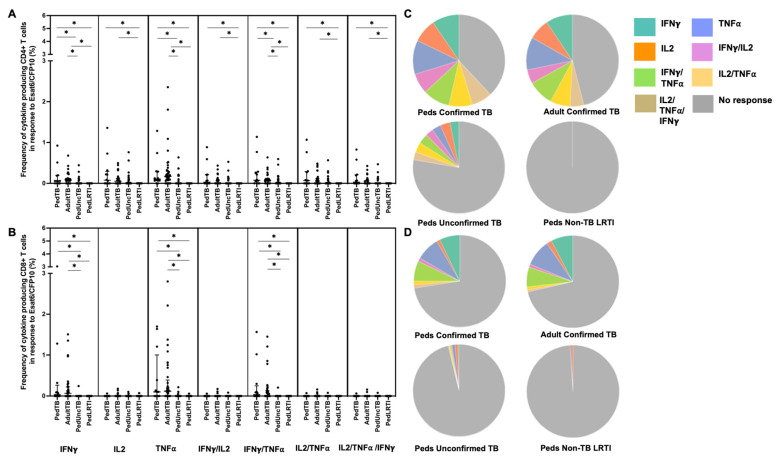
**Distribution of CD4+ and CD8+ T cells that produce cytokines in response to ESAT6/CFP10**. Median percentages (frequencies) of CD4+ (**A**) and CD8+ (**B**) T cells that produce IFN-γ, IL-2, or TNF-α in response to EC peptide pools were quantified by ICS (n = 12 PedTB; n = 41 AdultTB; n = 41 PedUncTB; n = 39 PedLRTI). Background correction (subtraction of the unstimulated/resting responses) was performed, and responses < 0.05% set to zero. Comparisons of medians among cohorts were performed using Kruskal–Wallis test; post hoc comparisons were performed using Dunn’s test of multiple comparisons. Shown are medians with IQR. Pie charts visualize the fraction of donors within each cohort with unresponsive CD4+ (**C**) and CD8+ T cells (**D**) (grey slices). The colored slices visualize the fraction of exclusively single, double, or triple positive CD4+ (**C**) or CD8+ (**D**) T cells among those participants with a detectable cytokine response within each cohort. * Indicates adjusted *p*-value < 0.05.

**Figure 3 pathogens-12-01353-f003:**
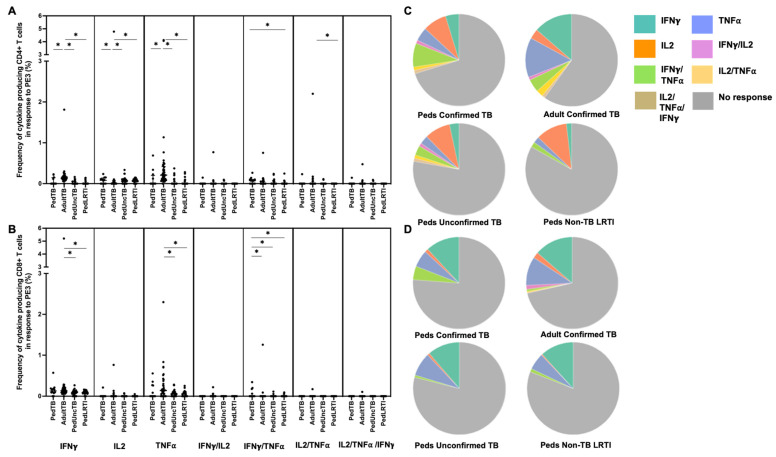
**Distribution of CD4+ and CD8+ T cells that produce cytokines in response to PE3.** Median percentages (frequencies) of CD4+ (**A**) and CD8+ (**B**) T cells that produce IFN-γ, IL-2, or TNF-α in response to PE3 peptide pool were quantified by ICS (n = 12 PedTB; n = 41 AdultTB; n = 38 PedUncTB; n = 39 PedLRTI). Background correction (subtraction of the unstimulated/resting responses) was performed, and responses <0.05% set to zero. Comparisons of medians among cohorts were performed using Kruskal–Wallis test; post hoc comparisons were performed using Dunn’s test of multiple comparisons. Shown are medians with IQR. Pie charts visualize the fraction of donors within each cohort with unresponsive CD4+ (**C**) and CD8+ T cells (**D**) (grey slices). The colored slices visualize the fraction of exclusively single, double, or triple positive CD4+ (**C**) or CD8+ (**D**) T cells among those participants with a detectable cytokine response within each cohort. * Indicates adjusted *p*-value < 0.05.

**Figure 4 pathogens-12-01353-f004:**
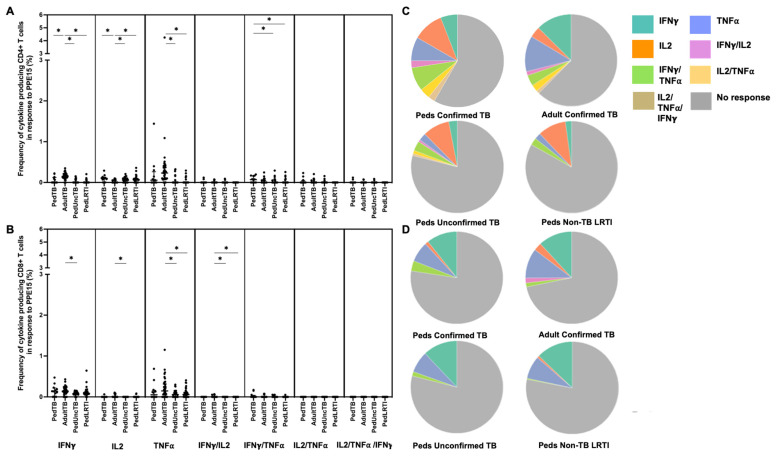
**Distribution of CD4+ and CD8+ T cells that produce cytokines in response to PPE15.** Median percentages (frequencies) of CD4+ (**A**) and CD8+ (**B**) T cells that produce IFN-γ, IL-2, or TNF-α in response to PPE15 peptide pool were quantified by ICS (n = 12 PedTB; n = 41 AdultTB; n = 38 PedUncTB; n = 39 PedLRTI). Background correction (subtraction of the unstimulated/resting responses) was performed, and responses <0.05% set to zero. Comparisons of medians among cohorts were performed using Kruskal–Wallis test; post hoc comparisons were performed using Dunn’s test of multiple comparisons. Shown are medians with IQR. Pie charts visualize the fraction of donors within each cohort with unresponsive CD4+ (**C**) and CD8+ T cells (**D**) (grey slices). The colored slices visualize the fraction of exclusively single, double, or triple positive CD4+ (**C**) or CD8+ (**D**) T cells among those participants with a detectable cytokine response within each cohort. * Indicates adjusted *p*-value < 0.05.

**Figure 5 pathogens-12-01353-f005:**
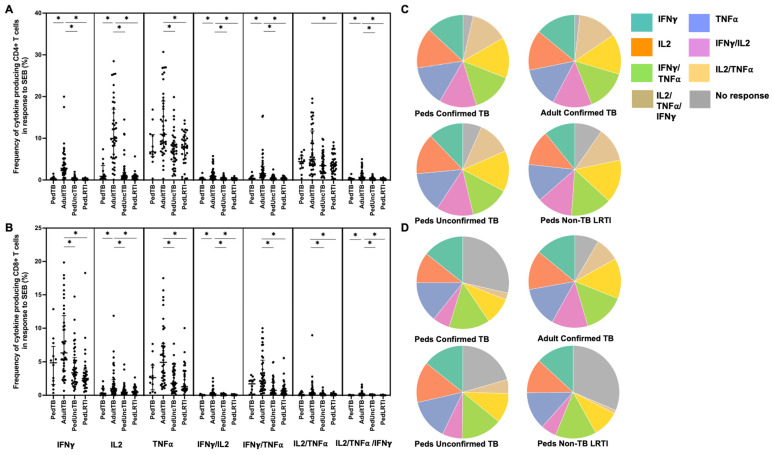
**Distribution of CD4+ and CD8+ T cells that produce cytokines in response to SEB.** Median percentages (frequencies) of CD4+ (**A**) and CD8+ (**B**) T cells that produce IFN-γ, IL-2, or TNF-α in response to SEB were quantified by ICS (n = 12 PedTB; n = 41 AdultTB; n = 41 PedUncTB; n = 39 PedLRTI). Background correction (subtraction of the unstimulated/resting responses) was performed, and responses <0.05% set to zero. Comparisons of medians among cohorts were performed using Kruskal–Wallis test; post hoc comparisons were performed using Dunn’s test of multiple comparisons. Shown are medians with IQR. Pie charts visualize the fraction of donors within each cohort with unresponsive CD4+ (**C**) and CD8+ T cells (**D**) (grey slices). The colored slices visualize the fraction of exclusively single, double, or triple positive CD4+ (**C**) or CD8+ (**D**) T cells among those participants with a detectable cytokine response within each cohort. * Indicates adjusted *p*-value < 0.05.

**Table 1 pathogens-12-01353-t001:** Antigens utilized in ICS assay.

Antigen	Rv Number
ESAT6/CFP10	Rv3875/Rv3874
EsxJ Family	Rv1038c, Rv1197, Rv1792, Rv2347, Rv3620c
PE12:PE13	Rv1172c(32):Rv1195(18)
PE3	Rv0159c(50)
PPE15	Rv1039c(50)
PPE51	Rv3136(46)

**Table 2 pathogens-12-01353-t002:** Demographic characteristics of study populations.

	Age (Median; IQR)	Sex (% Female)	BCG Scar Present	Weight-For-Age Z-Score (Median; IQR)	TST (mm) (Mean; SD)
**Confirmed pediatric TB (n = 12)**	30.8 months (28.17)	50%	33.3%	−1.55 (2.1)	16.3 (6.6)
**Unconfirmed pediatric TB (n = 41 *)**	18 months (30.9)	38.5%	69.2%	−1.76 (1.13)	11 (7.3)
**Non-TB pediatric LRTI (n = 39)**	12.3 months (13.1)	33%	87%	−1.30 (1.19)	0 (0)
**Adult confirmed TB (n = 41 *)**	23 years (8)	57.5%	Not collected	Not collected	Not collected

* Clinical data unavailable on 2/41 children with unconfirmed TB and 1/41 adults with confirmed TB; IQR—interquartile range; TST—tuberculin skin test; LRTI—lower respiratory tract infection; SD—standard deviation.

## Data Availability

Data and materials pertaining to this study are freely available upon request to the corresponding author.

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
