# Peer review of "Production of Proinflammatory Cytokines by CD4+ and CD8+ T Cells in Response to Mycobacterial Antigens among Children and Adults with Tuberculosis"

_pathogens, 2023, doi:10.3390/pathogens12111353_

Round 1

Reviewer 1 Report

Comments and Suggestions for Authors

Reviewer comment for Editor and authors

I appreciate the opportunity to review the manuscript entitled "Production of proinflammatory cytokines by CD4+ and CD8+ T 2 cells in response to mycobacterial antigens among children and 3 adults with tuberculosis". In the current study by Erin Morrow and colleagues have evaluated CD4+ and CD8+ T cell production of proinflammatory cytokines IFN-gamma, IL-2, and TNF-alpha in response to six mycobacterial antigens and superantigen staphylococcal enterotoxin B.

However, the author may take note of the major and minor remarks listed below to improve the manuscript:

I have major concerns about the study design and interpretation of the results.

Abstract:

The abstract is poorly written and does not provide a concise overview of the study's objectives and findings. The abstract should provide a clear summary of the study's objectives, methods, findings, and conclusion.

  • I kindly request that authors make it a little more elaborative for each section.

Introduction:

The introduction needs to be rewritten, as it seems out of context. It should effectively introduce the key concepts, research gaps, and objectives of the study.

Here are a few suggestions for improvement:

  • The introduction should briefly place the study in a broad context and highlight why it is important. It should define the purpose of the work and its significance, including the specific hypotheses being tested.
  • What are the major 'research gaps' that lead to the study hypothesis?

Material and methods:

This section should provide a solid foundation for conducting the study. However, in this article, some areas could be further improved to enhance clarity and reproducibility.

  • Please make sure that the methodology used is reproducible. It will be very good if authors can put it section-wise.
  • I would request that authors clarify the idea behind using staphylococcal enterotoxin B as a positive control. If possible, please include the data observed with mitogen-based stimulation conditions.
  • Also, to enhance clarity and reproducibility, please cite the used methodologies.

Results:

In this section, please rewrite the table and figure legends as they should be self-explanatory, including the sample number and appropriate statistical test used.

  • For example, plots showing serum levels of interleukin-1, soluble interleukin-2 receptor, interleukin-6, and tumor necrosis factor-β, the peripheral blood percentage of CD4+ T cells and CD8+ T cells, and the CD4+/CD8+ ratio in individuals with pulmonary tuberculosis according to smear grade Smear grades were classified as 1+, 2+, and 3+. Horizontal lines indicate the mean with SD.
  • Instead of adding intracellular flow cytometry as supplementary material, I would encourage authors to include this data for IFN-γ, IL-2, and TNF-α.
  • In addition to this, the data shown in supplementary figures 2, 3, and 4 is more representable in the form of a pie chart. I would recommend either changing the format of the pie to another or just putting it as shown in the supplementary figures.

Discussion:

I request authors rewrite this section.

  • The authors should discuss the results and how they can be interpreted from the perspective of previous studies and the working hypotheses. The findings and their implications should be discussed in the broadest context possible, and the limitations of the work should be highlighted. Future research directions may also be mentioned.
  • Correct me if I am wrong here. Based on the observed results, the study seems to be in line with the evaluation of the comparative immune response among children and adults. However, it looks like it derailed the vaccination and diagnostic approach.
  • In my opinion, the hypothesis being tested is not clear, and hence the manuscript requires major changes.

Reviewer 2 Report

Comments and Suggestions for Authors

Tuberculosis is a serious problem in many low and middle income countries. Currently, this pathology is the deadliest among infectious diseases. Children are especially vulnerable because their immunity is not fully developed. In your manuscript “Production of proinflammatory cytokines of CD4+ and CD8+ T-cells in response to mycobacterial antigens among children and adults with tuberculosis,” you attempted to evaluate the cytokine response of CD4+ and CD8+ T-cells to specific and nonspecific markers of Mycobacterium tuberculosis in children and adults. This work is important in the context of the global fight against tuberculosis, however, I have a large number of comments regarding the semantic part and design. You can check them out below.

Major concern

1) From a cellular perspective, there are two objects in your study: CD4+ and CD8+ T cells. In this case, according to the text of the manuscript you switch to the term Th-1 response. Of these two cells, only CD4+ are helpers, you wrote about this. Whereas CD8+ are cytotoxic T lymphocytes, killer T cells, and so on. And you haven’t written about this anywhere. This may cause some readers to be confused that CD8+ cells are also Th-1 cells. Still, Pathogens is a general infectious diseases journal, and not all readers are familiar with immunological terminology. Please describe the role of CD8+ cells in the pathogenesis of tuberculosis.

2) Your manuscript has a number of problems with pie charts. So, in the description of Figure 2 you write the following: “Pie charts visualize the median percentage of CD4+ (A) and CD8+ (B) T cells that produced different patterns of cytokines in response to EC-peptide pools among cohorts.” Further, you write that the data can be viewed in Supplemental Table 1 and Supplemental Figure 2. But if you compare the figure from the manuscript and Supplemental Table 1, a number of questions arise. For example, the “Peds Unconfirmed TB” group shows that some of their CD4+ cells produce IFN-γ, while Supplemental Table 1 says that for this group the median percentage (IQR) = 0.00 (0.00). The same is true for TNF-α and IL-2. In Supplemental Table 1 for all these cytokines in the “Peds Unconfirmed TB” group (for ESAT-6/CFP-10) median percentage (IQR) = 0.00 (0.00). However, in the pie chart, these cytokines have their own shaded sectors. The same applies to cytokines produced by CD8+ T cells. The pie chart in the “Peds Unconfirmed TB” group in Figure 2 has shaded sectors for the three cytokines listed above. However, in Supplemental Table 1 the median percentage (IQR) for them is also 0.00 (0.00).

3) Another comment relates directly to the design of pie charts. A pie chart displays 100% of a set. Individual sectors display the proportions of elements or groups of elements in this set. In a pie chart, three separate sectors cannot exist simultaneously: cells producing TNF-α; IL-2 producing cells and TNF-α/IL-2 producing cells. Because the set “cells producing TNF-α/IL-2” and the sets “cells producing TNF-α” and “cells producing IL-2” are overlapping sets by definition and, therefore, cannot be separate sectors in the pie chart. This comment applies to all other subsets of cells in the pie chart structure (cells that produce more than one cytokine). Due to the fact that you have overlapping sets marked as separate sectors, you get the impression that there are many more cytokine-producing cells than there actually are.

Minor concerns

Line 34: I would recommend that you add more keywords. Pathogens allows you to add from 3 to 10 keywords. This opportunity must not be missed. I advise you to add popular words like "cytokines" and "T-cells", as well as some others. However, this comment is advisory in nature. You can do as you please.

Line 130: Can you please describe in more detail the method of isolating PBMC or refer to some work where it is described in detail.

Line 131: Please write down the temperature at which the cells were preserved.

Figure 5: Your figures 5 (A and B) are beautiful, I like the color scheme and design choice, but they are very small. Could you make them bigger? As a reader, I would prefer to see these two figures separately from each other, so that they are as large as possible and easy to read.

And one more comment about Figure 5. The description of the figure says that the data is presented in the form of median and IQR. At the same time, there are no dots in the figure indicating individual indicators of patients above the third quartile limit. Did you remove these points from the figure in favor of the graph design? Or is it actually not the first and third quartiles, but the minimum and maximum of the sample? If the first option, then write about it in the description of the figure. If the second option, then correct this in the description of the figure.

Round 2

Reviewer 1 Report

Comments and Suggestions for Authors

Dear Team,

Thank you for adding all the necessary changes. The manuscript can be accepted in its present form.